# Efficient Adversarial Detection and Purification with Diffusion Models

## Abstract

Adversarial training and adversarial purification are two effective and practical defense methods to enhance a model's robustness against adversarial attacks. However, adversarial training necessitates additional training, while adversarial purification suffers from low time efficiency. More critically, current defenses are designed under the perturbation-based adversarial threat model, which is ineffective against the recently proposed unrestricted adversarial attacks. In this paper, we propose an effective and efficient adversarial defense method that counters both perturbation-based and unrestricted adversarial attacks. Our defense is inspired by the observation that adversarial attacks are typically located near the decision boundary and are sensitive to pixel changes. To address this, we introduce adversarial anti-aliasing to mitigate adversarial modifications. Additionally, we propose adversarial super-resolution, which leverages prior knowledge from clean datasets to benignly recover images. These approaches do not require additional training and are computationally efficient. Extensive experiments against both perturbation-based and unrestricted adversarial attacks demonstrate that our defense method outperforms state-of-the-art adversarial purification methods.

## 1 Introduction

Deep learning models have demonstrated remarkable performance across various tasks (He et al., 2016; Liu et al., 2021; Xiang et al., 2021). With the rapid advancement and widespread deployment of these models, their security and robustness are garnering increasing attention.

It is widely recognized that deep learning models are highly vulnerable to adversarial attacks (Madry et al., 2018; Carlini & Wagner, 2017). These attacks are performed by adding imperceptible perturbations to clean images. The perturbed images, known as adversarial examples, can deceive trained deep learning classifiers with high confidence while appearing natural and realistic to human observers. To mitigate adversarial attacks and ensure the stability of deep learning models, adversarial training (Madry et al., 2018; Gowal et al., 2021) has been developed. This approach aims to defend against adversarial attacks by training the classifier with adversarial examples. However, adversarial training tends to perform poorly against unknown attacks.

Recently, with the development of diffusion models (Dhariwal & Nichol, 2021; Rombach et al., 2022), adversarial purification (Nie et al., 2022; Song et al., 2024) has shown promising defense performance by recovering the adversarial examples to clean images. These works adopt the diffusion model's reverse generation process to gradually remove the Gaussian noise from the forward process and the adversarial perturbations. Nevertheless, these methods require heavy computational resources during the purification, which may not be practical in real-time scenarios.

Diffusion models also facilitate stronger unrestricted adversarial attacks (Chen et al., 2023b; Dai et al., 2023; Chen et al., 2023c). These unrestricted adversarial examples (UAEs) are generated through the reverse generation process by incorporating adversarial guidance. Unlike traditional perturbation-based adversarial attacks, UAEs exhibit superior attack performance against current defenses due to their distinct threat models. These attacks pose a new threat to the development of deep learning models and urgently need to be addressed. Even wrose, existing defenses have merely covered the discussion against UAEs.

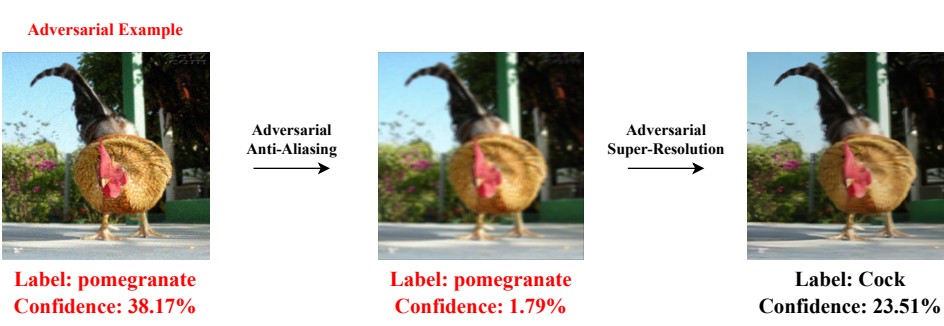

Figure 1: **The proposed adversarial defense pipeline.** We give an adversarial example of "cock" class with AutoAttack $\ell_{\inf} = 8/255$ on ImageNet dataset. Adversarial anti-aliasing aims to eliminate adversarial perturbations, while adversarial super-resolution seeks to restore benign images from blurred adversarial examples using prior knowledge from the clean dataset.

In this paper, we propose an effective adversarial defense method that detects both perturbation-based adversarial examples and unrestricted adversarial examples. To achieve the defense objective, we locate and utilize the common characteristic of these two types of attacks that both adversarial examples are generated close to the decision boundary for minimal perturbations, which makes these adversarial examples susceptible to changes in pixels.

Our defense employs zero-shot adversarial detection by extracting the "semantic shape" information from images without the image details, as illustrated in Figure 1. Specifically, we use adversarial anti-aliasing with specialized filters to blur the detailed adversarial modifications in the adversarial examples. Following this, we apply adversarial super-resolution to the anti-aliased adversarial examples, upscaling the blurred images using details from pre-trained clean super-resolution diffusion models. These two methods are time-efficient and do not require any modifications to the original models. To demonstrate the effectiveness of our proposed defense, we further validate its performance by using the upscaled adversarial examples as input for adversarial purification. Experiments on various datasets show that our defense outperforms state-of-the-art adversarial defenses in both adversarial detection and adversarial purification.

Our contributions are summarized as follows:

- We propose a novel adversarial defense capable of countering both perturbation-based adversarial examples and unrestricted adversarial examples, addressing the current gap in effective defenses against unrestricted adversarial attacks.

- We introduce various zero-shot and gradient-free defense strategies that preserve the semantic information of adversarial examples while eliminating adversarial modifications. These strategies include adversarial anti-aliasing for "semantic" extraction and adversarial super-resolution for incorporating benign priors and recovering benign details from adversarial examples.

- We conduct extensive experiments on various datasets against adaptive adversarial attacks. The results demonstrate the effectiveness of our proposed defense method compared to state-of-the-art adversarial defenses. Moreover, anti-aliased and upscaled adversarial examples effectively integrate with existing diffusion-based adversarial purification, validating the usability and scalability of our approach.

## 2 BACKGROUND

### 2.1 ADVERSARIAL TRAINING

Adversarial training (AT) is one of the most practical methods for enhancing a model's robustness against adversarial attacks. It involves training the model with both benign and adversarial data simultaneously during the training phase. However, robustness against unseen attacks remains a

significant challenge that affects the defense performance of traditional adversarial training (Madry et al., 2018). To address this, Gowal et al. (Gowal et al., 2021) and Rebuffi et al. (Rebuffi et al., 2021) have incorporated generated and augmented data to improve generalization by increasing data diversity. In addition to leveraging diverse data, refining the objective formulation of AT has also proven effective. By considering model weights, a wide range of adversarial training methods (Wu et al., 2020; Jin et al., 2023) have been proposed.

## 2.2 ADVERSARIAL PURIFICATION

Adversarial purification aims to eliminate adversarial perturbations in adversarial examples without requiring the re-training of deep learning models. These methods leverage the generative capabilities of generative models. Previous works utilizing generative adversarial networks (GANs) (Samangouei et al., 2018) and score-based matching models (Song et al., 2021; Yoon et al., 2021) have demonstrated state-of-the-art performance compared to adversarial training. With the advent of diffusion models, Nie et al. (Nie et al., 2022) discovered that diffusion-based adversarial purification methods outperform previous approaches in recovering clean images. However, finding the optimal generation steps for diffusion-based adversarial purification remains challenging. Additionally, adversarial images can negatively impact the reverse generation process of diffusion models. To address these issues, several works (Wang et al., 2022; Lee & Kim, 2023; Song et al., 2024) have proposed various solutions to enhance the performance of adversarial purification.

## 2.3 ADVERSARIAL EXAMPLE DETECTION

Adversarial example detection involves rejecting input data if it is identified as adversarial. These detection methods do not require re-training the classifier and do not modify clean data, making them particularly suitable for tasks that focus on data details. The most commonly discussed solution is to train a detector network specifically for adversarial detection. Existing approaches (Metzen et al., 2022; Yang et al., 2020) have employed various network architectures to train detectors, achieving satisfactory defense performance. Another detection method exploits the statistical divergence between benign and adversarial data. Grosse et al. (Grosse et al., 2017) and Song et al. (Song et al., 2018a) used different metrics to successfully identify adversarial examples within input data. Lastly, because adversarial examples are typically located near decision boundaries, their predictions are often inconsistent when input transformations are applied (Hu et al., 2019; Meng & Chen, 2017) or when the weights of the target models are altered (Feinman et al., 2017).

# 3 PRELIMINARY

## 3.1 THREAT MODEL

Adversarial examples conduct attacks by fooling the target model's classification result. Considering the untargeted attack scenario, the perturbation-based adversarial examples are defined as:

$$A_{\text{AE}} \triangleq \{x_{\text{adv}} = x + \delta | y \neq f(x), x \in D, |\delta| \leq \epsilon\} \tag{1}$$

where $\delta$ is the adversarial perturbation, $f(\cdot)$ is the target model, $D$ is the clean dataset, and $\epsilon$ is the perturbation norm constraint.

These adversarial examples are generated by adding the perturbations to the clean images. However, such perturbations can degenerate the image quality. By utilizing the generation models, Song et al. (Song et al., 2018b) presented unrestricted adversarial examples by directly generating adversarial examples with the generation tasks, which can be formulated as:

$$A_{\text{UAE}} \triangleq \{x_{\text{adv}} \in \mathcal{G}(z_{\text{adv}}, y) | y \neq f(x)\} \tag{2}$$

where $\mathcal{G}$ is the generation model, $z_{\text{adv}}$ is the latent code for generation.

These two adversarial examples are generated with different threat models. However, they both can successfully conduct attacks against the given target model. A robust defense method should be able to defend against these attacks simultaneously.

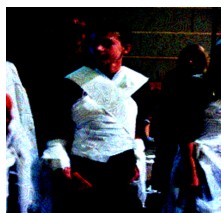 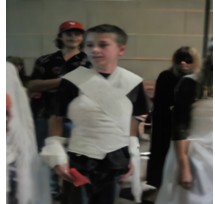

**AutoAttack Example**
**Robust Acc: 0%**

**RGB conversion**
**Robust Acc: 38.25%**

**Adv. Anti-Aliasing**
**Robust Acc: 55.85%**

Figure 2: **The vulnerability of adversarial examples to the changes in pixels.** AutoAttack can achieve nearly 100% attack success rate on the ImageNet dataset. However, with RGB conversions and image normalization, we can easily achieve around 38% robust accuracy. The proposed adversarial anti-aliasing is more effective while preserving the image quality.

### 3.2 DIFFUSION-BASED ADVERSARIAL PURIFICATION

The diffusion model (Ho et al., 2020) learns to recover the image from the denoising-like process, i.e., *reverse generation process*. The reverse generation process takes $T$ time steps to obtain a sequence of noisy data $\{x_{T-1}, \ldots, x_1\}$ and get the data $x_0$ at the last step. Specifically, it can be formulated as:

$$p_\theta(x_{t-1}|x_t) = \mathcal{N}(x_{t-1} : \mu_\theta(x_t, t), \Sigma_\theta(x_t, t)) \tag{3}$$

The *forward diffusion process* is where we iteratively add Gaussian noise to the data for training the diffusion model to learn $p_\theta(x_{t-1}|x_t)$. It is defined as:

$$q(x_t|x_{t-1}) = \mathcal{N}(x_t : \sqrt{\sigma_t}x_{t-1}, (1 - \sigma_t)\mathbf{I}) \tag{4}$$

where $\sigma$ is the noise schedule.

Nie et al. (Nie et al., 2022) attempted to find the optimal $t^*$ where it satisfy that:

$$\begin{aligned} x_{t^*} &= \sqrt{\sigma_{t^*}}x_{adv} + \sqrt{1 - \sigma_{t^*}}\varepsilon \\ &= \sqrt{\sigma_{t^*}}(x + \delta) + \sqrt{1 - \sigma_{t^*}}\varepsilon \end{aligned} \tag{5}$$

where $\varepsilon$ is the Gaussian noise $\varepsilon \sim \mathcal{N}(0, \mathbf{I})$. After we obtain the optimal $t^*$, we can utilize the reverse generation process over $x_{adv}$ to recover the clean $x$.

Wang et al. (Wang et al., 2022) utilized the whole reverse generation process with $T$ time step; they used adversarial sample $x_{adv}$ as guidance rather than an intermediate time step state. At each time step $t$, the guidance is added to the $x_t$ after the original reverse generation process and can be formulated as:

$$\nabla_x \log p(x_{adv}|x_t; t) = -R_t \nabla_{x_t} d(\hat{x}_t, x_{adv}) \tag{6}$$

where $R_t$ is the scale factor at $t$ time step, $d(\cdot)$ is the $\ell_2$ norm distance, and $\hat{x}_t$ is the estimation for $x_0$ at $t$ time step. The $\hat{x}_t$ is defined as:

$$\hat{x}_t = \frac{x_t - \sqrt{1 - \sigma_t}s_\theta(x_t)}{\sqrt{\sigma_t}} \tag{7}$$

where the $s_\theta$ known score function is defined as (Song et al., 2021).

## 4 METHODOLOGY

### 4.1 MOTIVATION

Despite the effectiveness of current adversarial defenses, such as adversarial training and adversarial purification, these methods require additional training and result in noticeable changes to the original images. These issues lead to low efficiency and can impact the original functionality of

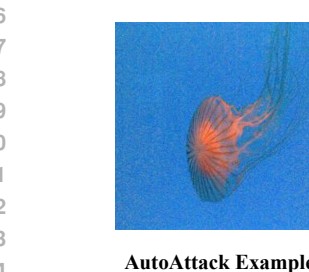
**AutoAttack Example**

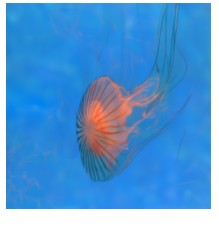
**MimicDiffusion**

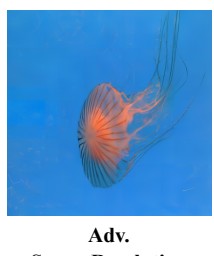
**Adv.
Super-Resolution**

Figure 3: **The example of proposed adversarial super-resolution.** Our method achieves similar adversarial purification without any gradient calculation of diffusion models.

deep learning models. To address these challenges, an effective defense that requires no additional training and makes no changes to clean images is needed to maintain the performance of the original models. Adversarial example detection is one of the most practical methods to meet these requirements. However, adversarial detection is often overlooked and has not been widely discussed in recent years. In this work, we propose an effective adversarial example detection method that achieves state-of-the-art defense performance without additional training or modifying the original images. Furthermore, we aim to defend against the recently proposed unrestricted adversarial attacks, which current defenses often ignore. To enhance the effectiveness of our defense, we also provide an adversarial purification method based on our adversarial example detection, offering a comprehensive discussion of adversarial defenses.

To achieve effective defenses against both unrestricted and perturbation-based adversarial attacks, it is essential to address their common characteristics. One critical factor is the value range of images: a valid RGB value is an integer between 0 and 255. However, the modifications introduced by various adversarial attacks are often performed using non-integer data types for gradient calculations. These modifications can become ineffective when transformed back to the RGB image format. Figure 2 supports our findings, showing that approximately 38% of adversarial examples from AutoAttack fail with simple RGB conversions. Furthermore, using these converted adversarial examples can enhance the performance of existing defenses. The reasons for this phenomenon could be that adversarial examples are typically located near the decision boundary and are sensitive to pixel changes. Therefore, our defense strategy focuses on finding effective conversions for adversarial examples to improve defense mechanisms.

### 4.2 ADVERSARIAL EXAMPLE DETECTION

Perturbation-based adversarial examples are precisely calculated based on the gradient of the loss function, whereas unrestricted adversarial examples are sampled near the decision boundary. Despite employing different threat models, both types of attacks produce adversarial examples that are sensitive to pixel changes. Since adversarial examples are designed to be imperceptible compared to clean images, the semantic shapes of objects within the images should correspond to their original labels. Therefore, our defense strategy focuses on extracting the semantic shapes from the adversarial examples and eliminating the adversarial pixel-level details.

#### 4.2.1 ADVERSARIAL ANTI-ALIASING

Anti-aliasing is a straightforward, zero-shot method for smoothing image details. Its effectiveness in adversarial defense has been demonstrated in recent research (Liang et al., 2018; Vasconcelos et al., 2021). Unlike previous works, we have found that anti-aliasing with non-square filters is particularly effective against adversarial attacks while preserving clean accuracy. Additionally, using the average value from neighboring pixels, excluding the original pixel, has also proven effective. This is because adversarial perturbations are calculated on a pixel-wise basis and are sensitive to pixel changes. Even with simple anti-aliasing, we achieve moderate defense performance, underscoring the effectiveness of our approach. To maintain the resolution of the output image, we use padding,

Table 1: **The defense performance against AutoAttack ($\ell_{\text{inf}} = 8/255$) on the CIFAR10 dataset.**

| Method | Target Model | Standard Accuracy(%) | Robust Accuracy(%) |
|---|---|---|---|
| Wu *et al.* Wu et al. (2020) | WideResNet-28-10 | 85.36 | 59.18 |
| Gowal *et al.* Gowal et al. (2021) | WideResNet-28-10 | 87.33 | 61.72 |
| Rebuffi *et al.* Rebuffi et al. (2021) | WideResNet-28-10 | 87.50 | 65.24 |
| Wang *et al.* Wang et al. (2022) | WideResNet-28-10 | 84.85 | 71.18 |
| Nie *et al.* Nie et al. (2022) | WideResNet-28-10 | 89.23 | 71.03 |
| Song *et al.* (Song et al., 2024) | WideResNet-28-10 | 92.10 | 75.45 |
| Ours$_{\text{Detection}}$ | WideResNet-28-10 | *97.50 $\pm$ 2.15* | *93.66 $\pm$ 0.42* |
| Ours$_{\text{Purification}}$ | WideResNet-28-10 | **92.54 $\pm$ 1.66** | **82.02 $\pm$ 1.17** |
| Rebuffi *et al.* Rebuffi et al. (2021) | WideResNet-70-16 | 88.54 | 64.46 |
| Gowal *et al.* Gowal et al. (2021) | WideResNet-70-16 | 88.74 | 66.60 |
| Nie *et al.* Nie et al. (2022) | WideResNet-70-16 | 91.04 | 71.84 |
| Song *et al.* (Song et al., 2024) | WideResNet-70-16 | 93.25 | 76.60 |
| Ours$_{\text{Detection}}$ | WideResNet-70-16 | *98.13 $\pm$ 1.94* | *93.66 $\pm$ 2.42* |
| Ours$_{\text{Purification}}$ | WideResNet-70-16 | **93.42 $\pm$ 1.51** | **83.65 $\pm$ 2.90** |

which is calculated as follows:

$$R_{out} = \lfloor R_{in} + 2 \times \text{Padding} - \text{filter\_size} \rfloor \tag{8}$$

where $R$ is the shape of the data. We use stride $= 1$.

### 4.2.2 ADVERSARIAL SUPER-RESOLUTION

During the adversarial anti-aliasing phase, we significantly reduce adversarial perturbations by directly decreasing the pixel-wise modifications of the adversarial examples. However, this approach may not be effective against unrestricted adversarial examples, as they are not generated by adding explicit perturbations. Additionally, blurring the images can negatively impact the clean accuracy of the target model. Super-resolution offers an effective way to recover high-quality images from our adversarial anti-aliased images. Previous super-resolution methods (Ledig et al., 2017; Gao & Zhuang, 2019) typically modify the original pixels of the low-resolution image and use the residual features of the original low-resolution image. These methods can inadvertently transfer negative effects from the adversarial examples to the final high-resolution images, making them ineffective for adversarial super-resolution. Diffusion-model-based super-resolution (Yue et al., 2024; Rombach et al., 2022) provides a more isolated approach to achieving super-resolution. These models generate high-resolution images through a denoising-like process over randomly sampled noise, using the low-resolution image as a condition.

In this work, we adopt the ResShift method by Yue et al. (Yue et al., 2024) for our super-resolution process. This super-resolution model can also incorporate benign priors for defense, as it is trained with the clean dataset of the target model. Figure 3 demonstrates that the proposed super-resolution method achieves results comparable to diffusion-based adversarial purification Song et al. (2024), which do not require calculation of gradient.

### 4.2.3 ADVERSARIAL DETECTION

The proposed adversarial detection method relies on the consistency of classification results between the input image and the image after adversarial super-resolution. Compared to existing adversarial training and adversarial purification methods, our adversarial detection achieves stronger defenses with higher robust accuracy. Additionally, our approach does not require any training of the target model or the defense model. Moreover, diffusion-model-based super-resolution requires significantly fewer diffusion time steps than diffusion-based adversarial purification.

$$y = \{f(\text{SR}(\text{AA}(x)))|f(x) = f(\text{SR}(\text{AA}(x)))\} \tag{9}$$

Table 2: **The defense performance against BPDA+EOT ($\ell_{inf} = 8/255$) on the CIFAR10 dataset with WideResNet-28-10 as the target model.**

| Method | Purification | Standard Accuracy(%) | Robust Accuracy(%) |
|---|---|---|---|
| Nie *et al.* Nie et al. (2022)($t^* = 0.0075$) | Diffusion | 91.38 | 77.62 |
| Nie *et al.* Nie et al. (2022)($t^* = 0.1$) | Diffusion | 89.23 | **81.56** |
| Wang *et al.* Wang et al. (2022) | Diffusion | 90.36 | 77.31 |
| Song *et al.* (Song et al., 2024) | Diffusion | 91.41 | 76.45 |
| Ours$_{\text{Detection}}$ | Diffusion | *97.55 ± 2.84* | *93.45 ± 0.84* |
| Ours$_{\text{Purification}}$ | Diffusion | **91.52 ± 1.28** | 81.24 ± 2.51 |

Table 3: **The defense performance against AdvDiff on the CIFAR10 dataset.**

| Method | Target Model | Standard Accuracy(%) | Robust Accuracy(%) |
|---|---|---|---|
| Nie *et al.* (Nie et al., 2022) | WideResNet-28-10 | 95.42 | 21.56 |
| Song *et al.* (Song et al., 2024) | WideResNet-28-10 | 96.21 | 23.23 |
| Ours$_{\text{Detection}}$ | WideResNet-28-10 | *96.80 ± 1.14* | *72.32 ± 3.45* |
| Ours$_{\text{Purification}}$ | WideResNet-28-10 | **96.80 ± 0.37** | **33.97 ± 0.77** |

#### 4.2.4 ADVERSARIAL PURIFICATION

To demonstrate the effectiveness of the proposed defense and provide a fair comparison with previous works, we further evaluate the adversarial purification performance on the adversarial examples after detection. Our adversarial purification leverages the generative capabilities of diffusion models.

## 5 EXPERIMENTS

### 5.1 EXPERIMENTAL SETUP

**Dataset and target models.** We consider CIFAR-10 (Krizhevsky et al., 2009) and ImageNet (Deng et al., 2009) for major evaluation. For target models, we adopt WideResNet-28-10 and WideResNet-70-16 (Zagoruyko & Komodakis, 2016) for CIFAR-10 dataset and ResNet50 (He et al., 2016) for ImageNet dataset. These are commonly adopted backbones for adversarial robustness evaluation.

**Comparisons.** We compared our defense methods with various state-of-the-art defenses by the standardized benchmark: RobustBench (Croce et al., 2021). We mainly compare two diffusion-based adversarial purification methods: Nie et al.'s DiffPure (Nie et al., 2022) and Song et al.'s MimicDiffusion (Song et al., 2024). We use the Score SDE Song et al. (2021) implementation of MimicDiffusion on CIFAR-10 for fair comparisons. The defense methods that use extra data are not compared for fairness. We only evaluate the adversarial purification methods against unrestricted adversarial attacks as the adversarial training's different threat model.

**Attack settings.** We evaluate our method with both perturbation-based attacks and diffusion-based unrestricted adversarial attacks. For perturbation-based attacks, we select AutoAttack (Croce & Hein, 2020), PGD (Madry et al., 2018). For diffusion-based unrestricted adversarial attacks, we use DiffAttack (Chen et al., 2023a) and AdvDiff (Dai et al., 2023) for comparisons. DiffAttack is only evaluated on the ImageNet dataset according to the original paper. To ensure a fair comparison with previous diffusion-based adversarial purification, we include the evaluation against the adaptive attack, i.e., Backward pass differentiable approximation (BPDA+EOT) (Hill et al., 2021). On CIFAR-10, the attack settings follow DiffPure (Nie et al., 2022). On ImageNet, we randomly sample 5 images from each class and average over 10 runs.

**Implementation details.** We use Ours$_{\text{Detection}}$ to represent adversarial detection. We adopt the mean filter with $[[1, 1], [1, 1]]$ for adversarial anti-aliasing on CIFAR-10, and $[[1, 1, 1, 1, 1], [1, 1, 0, 1, 1], [1, 1, 1, 1, 1]]$ in ImageNet. ResShift (Yue et al., 2024) is utilized for adversarial super-resolution. We implement the adversarial purification, noted as Ours$_{\text{Purification}}$, by

Table 4: **The defense performance against AutoAttack ($\ell_{\mathbf{inf}} = 8/255$) on the ImageNet dataset.**

| Method | Target Model | Standard Accuracy(%) | Robust Accuracy(%) |
|---|---|---|---|
| Engstrom *et al.* Croce et al. (2021) | ResNet50 | 62.56 | 31.06 |
| Wong *et al.* Wong et al. (2020) | ResNet50 | 55.62 | 26.95 |
| Salman *et al.* Salman et al. (2020) | ResNet50 | 64.02 | 37.89 |
| Bai *et al.* Bai et al. (2021) | ResNet50 | 67.38 | 35.51 |
| Nie *et al.* Nie et al. (2022) | ResNet50 | 68.22 | 43.89 |
| Song *et al.* (Song et al., 2024) | ResNet50 | 66.92 | 61.53 |
| Ours$_{\text{Detection}}$ | ResNet50 | *88.30 ± 2.44* | *83.14 ± 1.82* |
| Ours$_{\text{Purification}}$ | ResNet50 | **75.28 ± 1.06** | **67.61 ± 1.95** |

Table 5: **The defense performance against PGD ($\ell_{\mathbf{inf}} = 4/255$) on the ImageNet dataset.**

| Method | Target Model | Standard Accuracy(%) | Robust Accuracy(%) |
|---|---|---|---|
| Wong *et al.* Wong et al. (2020) | ResNet50 | 55.62 | 26.24 |
| Salman *et al.* Salman et al. (2020) | ResNet50 | 64.02 | 34.96 |
| Bai *et al.* Bai et al. (2021) | ResNet50 | 67.38 | 40.27 |
| Nie *et al.* Nie et al. (2022) | ResNet50 | 68.22 | 42.88 |
| Wang *et al.* Wang et al. (2022) | ResNet50 | 70.17 | 68.78 |
| Song *et al.* (Song et al., 2024) | ResNet50 | 66.92 | 62.16 |
| Ours$_{\text{Detection}}$ | ResNet50 | *88.30 ± 2.44* | *80.21 ± 2.50* |
| Ours$_{\text{Purification}}$ | ResNet50 | **75.28 ± 1.06** | **69.75 ± 2.61** |

the adversarial examples after the proposed upscale method. We use the official Score SDE Song et al. (2021) checkpoint for CIFAR-10 and LDM Rombach et al. (2022) checkpoint for ImageNet to generate UAEs. More details and experiment results are given in the appendix.

**Evaluation metrics.** Following Nie et al. (Nie et al., 2022), we use *standard accuracy* and *robust accuracy* as the evaluation metrics. Both are calculated according to the top-1 classification accuracy. To evaluate the proposed detection method, i.e., Ours$_{\text{Detection}}$, we report the detection accuracy of our detection methods over the data that passes the detection. For standard accuracy, we evaluate the number of clean images that **NOT** detected by our method, while we report the number of adversarial images that **DO** detected by our method for robust accuracy.

## 5.2 ATTACK PERFORMANCE

### 5.2.1 CIFAR10

**Perturbation-based adversarial attack**. Table 1 presents the defense performance against AutoAttack ($\ell_{\text{inf}} = 8/255$) on the CIFAR10 dataset. The results demonstrate that our proposed method achieves better standard accuracy and robust accuracy than previous attack methods. Our detection method achieves over a 90% detection rate against adversarial examples, indicating further improvements in our purification method. Because images in the CIFAR10 dataset are only with $32 \times 32$ resolution, we set our anti-aliasing filter to a relatively small size. Table 2 indicates that the robustness performance of the proposed method is on par with the state-of-the-art method (Nie et al., 2022). However, we can further enhance our performance by incorporating adversarial purification techniques from previous work. This finding suggests that our method is more suitable for high-resolution images, as $32 \times 32$ may not be large enough to effectively extract the semantic shape for our approach.

**Unrestricted adversarial attack**. Unrestricted adversarial examples on the CIFAR10 dataset are challenging to detect and defend against, as shown in Table 3. Our purification method outperforms the previous adversarial purification approach Song et al. (2024) by an average of 10%, validating the effectiveness of our proposed defense.

Table 6: **The defense performance against AdvDiff ($\ell_{inf} = 8/255$) on the ImageNet dataset.**

| Method | Target Model | Standard Accuracy(%) | Robust Accuracy(%) |
|---|---|---|---|
| Nie *et al.* Nie et al. (2022) | ResNet50 | 91.48 | 24.82 |
| Wang *et al.* Wang et al. (2022) | ResNet50 | 92.31 | 26.74 |
| Song *et al.* (Song et al., 2024) | ResNet50 | 92.54 | 25.35 |
| Ours$_{\text{Detection}}$ | ResNet50 | *92.10 ± 2.32* | *82.45 ± 4.65* |
| Ours$_{\text{Purification}}$ | ResNet50 | **97.83 ± 1.36** | **42.21 ± 3.41** |

### 5.2.2 IMAGENET

**Perturbation-based adversarial attack**. Tables 4 and 5 demonstrate that the proposed defense method achieves significantly higher performance in both standard accuracy and robust accuracy. Our defense's standard accuracy notably surpasses previous work, further validating that adversarial super-resolution effectively leverages prior knowledge from the training dataset to achieve better classification accuracy. Adversarial anti-aliasing proves to be particularly effective on the ImageNet dataset, where the filter successfully blurs adversarial perturbations in the detailed pixels of adversarial examples. Additionally, our adversarial detection method achieves approximately 85% detection performance on adversarial examples and only a 10% detection error on clean images, making it suitable for real-world applications and providing a foundation for further improvements in future defenses.

**Unrestricted adversarial attack**. We present the defense performance of various methods against the unrestricted adversarial attack AdvDiff in Table 6. The results indicate that current defenses are ineffective against the recently proposed unrestricted adversarial attacks. The high standard accuracy can be attributed to the strong generative performance of benign diffusion models. Our defense method is capable of detecting the majority of unrestricted adversarial examples and achieves significantly higher robust accuracy compared to previous defenses.

Table 7: **The average time cost of defending one image against PGD ($\ell_{inf} = 4/255$) on the ImageNet dataset.**

| Method | Defend Method | Time Cost(s) | Robust Accuracy(%) |
|---|---|---|---|
| Nie *et al.* Nie et al. (2022) | Diffusion | 13.3 | 42.88 |
| Wang *et al.* Wang et al. (2022) | Diffusion | 224 | 68.78 |
| Song *et al.* (Song et al., 2024) | Diffusion | 146 | 62.16 |
| Ours | Adversarial Anti-Aliasing | $3e^{-3}$ | 57.61 |
| + | Adversarial Super-Resolution | **1.1** | **69.62** |

### 5.3 TIME EFFICIENCY

We evaluate the average time for defending against one adversarial example as shown in Table 7. The results indicate that our proposed method achieves better robust accuracy with significantly lower time costs, as it does not require any gradient calculations over the diffusion model. Notably, our adversarial anti-aliasing can defend against approximately 57% of adversarial examples in just $3e^{-3}$ seconds. Furthermore, we can enhance the defense performance of our method by combining it with previous purification methods, with only a minimal tradeoff in time cost.

### 5.4 ABLATION STUDY

We perform ablation studies to validate the performance of the proposed detection methods. We evaluate the defense method against AutoAttack ($\ell_{inf} = 8/255$) on the ImageNet dataset by default.

**Adversarial Anti-Aliasing**. Despite the satisfactory robustness performance of the proposed adversarial anti-aliasing, the choice of filter settings is critical for optimal defense performance. We

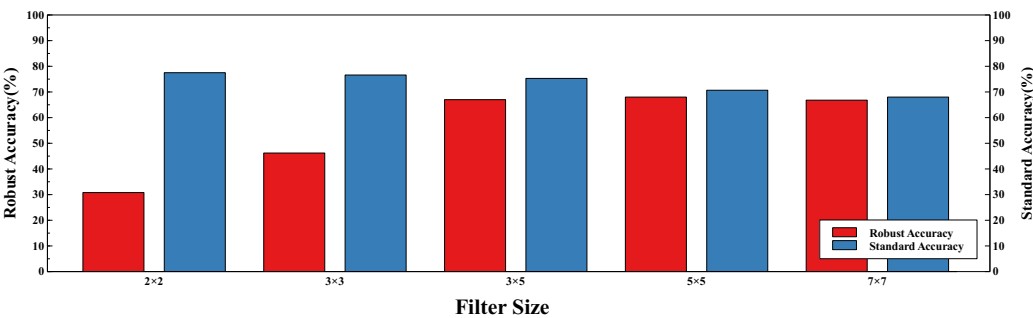

Figure 4: **The ablation study of filter size.**

| Method | Robust Accuracy(%) |
|---|---|
| Nie *et al.* Nie et al. (2022) | 43.89 |
| Song *et al.* (Song et al., 2024) | 61.53 |
| Adversarial AA | 55.85 |
| Adversarial SR | 41.23 |
| Adversarial AA+SR | **67.01** |

| Method | Robust Accuracy(%) |
|---|---|
| Nie *et al.* Nie et al. (2022) + Ours | 43.89 69.44 |
| Song *et al.* (Song et al., 2024) + Ours | 61.53 72.18 |

(a) **The ablation study of proposed adversarial super-resolution.**

(b) **The performance of integrating our method with previous adversarial purification.**

present the defense performance with different filters in Figure [reference]. The results indicate a tradeoff between robust accuracy and standard accuracy. Robust accuracy tends to stabilize when using a filter larger than $3 \times 3$ in size. Therefore, it is relatively straightforward to identify a suitable filter with a few attempts. Furthermore, the filter settings are generalized across different adversarial attacks within the same dataset, as demonstrated in Tables 4, 5, and 6.

**Adversarial Super-Resolution**. The proposed adversarial super-resolution achieves a similar purification function to previous diffusion-based adversarial purification methods, but without the need for computationally expensive gradient calculations. Table 8a demonstrates that our method slightly outperforms traditional adversarial purification when using anti-aliased adversarial examples as input. However, it is crucial to use anti-aliased adversarial examples for optimal performance in adversarial super-resolution, as we do not account for the adversarial gradient during the super-resolution process.

**Adversarial Purification**. We can enhance diffusion-based adversarial purification methods from previous works by replacing the adversarial input with the adversarial examples after detection. The processed adversarial examples are more benign and closer to the clean images, thereby enabling better purification performance, as demonstrated in Table 8b.

## 6 CONCLUSION

In this paper, we present an effective and efficient adversarial defense method against both perturbation-based and unrestricted adversarial attacks. The proposed techniques, adversarial anti-aliasing and adversarial super-resolution, effectively eliminate adversarial modifications and recover benign images with minimal computational overhead. Comprehensive experiments on the CIFAR-10 and ImageNet datasets validate that our proposed defense outperforms state-of-the-art defense methods. Our work demonstrates that simple adversarial anti-aliasing can achieve moderate model robustness with almost no additional cost. Furthermore, the proposed super-resolution method can perform adversarial purification without requiring the calculation of the diffusion model's gradient. We hope our work will serve as a baseline for the further development of adversarial defenses.

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
