# Supplementary Materials for Efficient Adversarial Detection and Purification with Diffusion Models

## A  Detail Experiment Settings

Our experiment is implemented with PyTorch on an NVIDIA GeForce RTX 3090 GPU. For the CIFAR10 dataset, we upscale the adversarial anti-aliased images with PyToch to $64 \times 64$ resolution for ResShift. We use ResShift default v3 parameter for our experiments.

## B  Additional Experiments

We further report additional experiments against various adversaries on CIFAR10 and ImageNet datasets, as shown in Table 1 and 2. Noted that DiffAttack Chen et al. (2023) adopt latent inversion from the validation set to generate adversarial examples, we only report the standard accuracy to the clean validation set.

Table 1: **The defense performance against AutoAttack ($\ell_2 = 0.5$) on the CIFAR10 dataset.**

| Method | Target Model | Standard Accuracy(%) | Robust Accuracy(%) |
|---|---|---|---|
| Rony *et al.* (Rony et al., 2019) | WideResNet-28-10 | 89.05 | 66.41 |
| Ding *et al.* (Ding et al., 2020) | WideResNet-28-10 | 88.02 | 67.77 |
| Rebuffi *et al.* (Rebuffi et al., 2021) | WideResNet-28-10 | 91.79 | 78.32 |
| Wang *et al.* (Wang et al., 2022) | WideResNet-28-10 | 92.00 | 75.28 |
| Nie *et al.* (Nie et al., 2022) | WideResNet-28-10 | 91.38 | 78.98 |
| Song *et al.* (Song et al., 2024) | WideResNet-28-10 | **92.84** | 81.52 |
| Ours$_{\text{Detection}}$ | WideResNet-28-10 | *97.50 ± 2.15* | *92.95 ± 0.36* |
| Ours$_{\text{Purification}}$ | WideResNet-28-10 | 92.54 ± 1.66 | **84.90 ± 2.82** |
| Gowal *et al.* (Gowal et al., 2021) | WideResNet-70-16 | 90.90 | 74.03 |
| Rebuffi *et al.* (Rebuffi et al., 2021) | WideResNet-70-16 | 92.41 | 80.86 |
| Nie *et al.* (Nie et al., 2022) | WideResNet-70-16 | 93.24 | 81.17 |
| Song *et al.* (Song et al., 2024) | WideResNet-70-16 | 92.51 | 83.60 |
| Ours$_{\text{Detection}}$ | WideResNet-70-16 | *98.13 ± 1.94* | *94.57 ± 1.82* |
| Ours$_{\text{Purification}}$ | WideResNet-70-16 | **93.42 ± 1.51** | **87.60 ± 2.35** |

Table 2: **The defense performance against DiffAttack ($\ell_{\text{inf}} = 8/255$) on the ImageNet dataset.**

| Method | Target Model | Standard Accuracy(%) | Robust Accuracy(%) |
|---|---|---|---|
| Nie *et al.* Nie et al. (2022) | ResNet50 | 68.22 | 59.15 |
| Song *et al.* (Song et al., 2024) | ResNet50 | 66.92 | 60.17 |
| Ours$_{\text{Detection}}$ | ResNet50 | *88.30 ± 2.44* | *78.44 ± 1.95* |
| Ours$_{\text{Purification}}$ | ResNet50 | **75.28 ± 1.06** | **65.51 ± 1.33** |

| 1 | 1 | 1 | 1 | 1 |
|---|---|---|---|---|
| 1 | 1 | 1 | 1 | 1 |
| 1 | 1 | 0 | 1 | 1 |
| 1 | 1 | 1 | 1 | 1 |
| 1 | 1 | 1 | 1 | 1 |

**Robust Acc: 68%**

| 1 | 1 | 1 | 1 | 1 |
|---|---|---|---|---|
| 1 | 1 | 1 | 1 | 1 |
| 1 | 1 | 1 | 1 | 1 |
| 1 | 1 | 1 | 1 | 1 |
| 1 | 1 | 1 | 1 | 1 |

**Robust Acc: 60%**

| 0 | 0 | 1 | 0 | 0 |
|---|---|---|---|---|
| 0 | 1 | 1 | 1 | 0 |
| 1 | 1 | 0 | 1 | 1 |
| 0 | 1 | 1 | 1 | 0 |
| 0 | 0 | 1 | 0 | 0 |

**Robust Acc: 57%**

| 0 | 0 | 1 | 0 | 0 |
|---|---|---|---|---|
| 0 | 0 | 1 | 0 | 0 |
| 1 | 1 | 0 | 1 | 1 |
| 0 | 0 | 1 | 0 | 0 |
| 0 | 0 | 1 | 0 | 0 |

**Robust Acc: 55%**

Figure 1: **The defense performance of various filter weights against AutoAttack ($\ell_{\text{inf}} = 8/255$) on the ImageNet dataset.** We use 1 for better understanding, while we set to the mean value according to the number of 1 blocks in the experiments.

## C   FILTER SELECTION

We discuss the choice of filter settings in the ablation study. However, it is also critical to design the filter weight for the adversarial anti-aliasing. Figure 1 demonstrates that the selection of filter weight is empirical and achieves the best performance on setting the mean value except for the center.

## D   LIMITATION

Despite achieving a significantly higher time efficiency and better defense performance than previous diffusion-based adversarial purification, our defense still has several limitations. One drawback is there exists a gap between the adversarial detection rate and robust accuracy. Therefore, a stronger defense can be proposed to increase the robust accuracy that focuses on defending against the detected adversarial examples. Another drawback is that the robust accuracy against UAEs is still not comparable to perturbation-based adversarial attacks. We aim to further improve it in future work.