# OpenReview forum: "Efficient Adversarial Detection and Purification with Diffusion Models"
_ICLR.cc/2025/Conference — ICLR 2025 Conference Withdrawn Submission_

### Official Review · Reviewer_iNqp · 2024-11-01

**Soundness:** 2
**Presentation:** 1
**Contribution:** 2
**Rating:** 3
**Confidence:** 4

**Summary:**

This paper introduces an adversarial detection and purification method that utilizes a diffusion model without additional training, designed to defend against both perturbation-based and unrestricted adversarial attacks. The experiments conducted on CIFAR-10 and ImageNet datasets demonstrate enhanced robustness and defense efficiency.

**Strengths:**

- The defense method, including anti-aliasing and super-resolution, can defend against both perturbation-based and unrestricted adversarial attacks.
- The defense method demonstrates higher defensive efficiency.

**Weaknesses:**

- The **presentation needs improvement**. The title of section 4.2 is "Adversarial Example Detection," yet within this section, subsection 4.2.3 is titled "Adversarial Detection," and subsection 4.2.4 is titled "Adversarial Purification." There is a logical disorganization between the sections.
- The paper **combines both detection and purification methods**. It is uncertain whether there is a clear enhanced performance over previous works when considering either detection or purification alone.
- **Lack of novelty**; the methods of anti-aliasing and super-resolution are somewhat trivial, and there is a lack of strategies to enhance defensive efficiency, which is the main proposal in the title.
- The effectiveness of a standalone purification method without detection is questionable. According to my understanding, this paper does not improve the purification method. If there is a misunderstanding here, please clarify the specific differences between your purification method and previous works.

**Questions:**

please see weaknesses

---

### Official Review · Reviewer_SkoJ · 2024-11-02

**Soundness:** 4
**Presentation:** 4
**Contribution:** 3
**Rating:** 5
**Confidence:** 3

**Summary:**

Current defenses are primarily designed for perturbation-based adversarial threat models, rendering them ineffective against recently proposed unrestricted adversarial attacks. In this paper, the authors introduce an effective and efficient adversarial defense method that addresses both perturbation-based and unrestricted attacks. This defense is inspired by the observation that adversarial attacks are typically located near the decision boundary and are sensitive to pixel alterations. To counter this, they introduce adversarial anti-aliasing to reduce adversarial modifications. Additionally, they propose adversarial super-resolution, which utilizes prior knowledge from clean datasets to recover images in a benign manner. These approaches do not require additional training. Extensive experiments against both perturbation-based and unrestricted adversarial attacks demonstrate that the proposed defense method outperforms state-of-the-art adversarial purification techniques.

**Strengths:**

1.	The proposed method does not require any additional training.
2.	The paper is well-written, and the proposed method is reproducible.
3.	The research content holds practical value.
4.	The proposed method has been tested against several adversarial techniques and shows a clear defensive effect.

**Weaknesses:**

1.	The paper's innovation is insufficient; the method proposed by the authors resembles a combination of existing approaches.
2.	Although the paper compares the proposed method with existing techniques, the analysis of differences between these methods is insufficient, particularly regarding performance variations under different attack types.

**Questions:**

1.	Although the method proposed by the authors does not require training, the use of generative models necessitates further analysis of its efficiency to enhance comparisons with mainstream defense methods.  Can you give some analysis or explanation?

---

### Official Review · Reviewer_LgMt · 2024-11-02

**Soundness:** 1
**Presentation:** 2
**Contribution:** 2
**Rating:** 3
**Confidence:** 4

**Summary:**

This paper proposes a detection and purification method for adversarial defense. The method is motivated by the observation that the effectiveness of adversarial examples is vulnerable to small pixel changes. To achieve adversarial purification, an antialiasing step is applied to the input image, followed by a super-resolution step using the diffusion-based ResShift model. Adversarial detection is implemented by examining whether the raw sample and the purified sample yield the same model output. Experiments on CIFAR10 and ImageNet suggest the effectiveness and efficiency of the proposed method against norm-constrained attacks and unrestricted attacks.

**Strengths:**

- It is pointed out that a significant proportion of adversarial images produced by AutoAttack can be deactivated by transforming them to valid integer RGB values, which suggests a potential flaw in existing robustness evaluation protocols, since a practical model typically accepts only RGB images with integer values.
- This paper considers unrestricted attacks in the experiments, which are not well-studied for adversarial purification methods.

**Weaknesses:**

- The visualization of the RGB conversion result in Figure 2 seems strange according to the statements in Lines 242-246, where rounding the RGB values of the AutoAttack example to integer and clipping them to 0-255 should not produce a significant variation.
- As a major technical contribution of the proposed method, the implementation of adversarial anti-aliasing is not clearly stated in Sec. 4.2.1.
- The attacks used in the experiments may be insufficient to assess the robustness of the proposed method. Specifically, it has been suggested by (Lee & Kim, 2023) that the AutoAttack and BPDA used in this paper tend to overestimate the robustness of diffusion-based purification methods. Instead, PGD+EOT with exact gradients of the complete computation graph (i.e., including the proposed adversarial AA+SR) should be the more reliable adaptive attack. This may also apply to the unrestricted attacks.

**Questions:**

- How is the "RGB conversion" in Figure 2 implemented?

---

### Official Review · Reviewer_RJds · 2024-11-03

**Soundness:** 1
**Presentation:** 3
**Contribution:** 1
**Rating:** 3
**Confidence:** 5

**Summary:**

This work presents a way to detect adversarial examples. The detection is based on the difference in the outputs of classifiers for the original image and the image that has gone through anti-aliasing and then super-resolution. Experiments are conducted on CIFAR-10 and ImageNet, and the results are compared with those of adversarial training and purification methods.

**Strengths:**

1.	The paper is well-written, and the illustrations clearly show the concepts in this work.
2.	The experiments are conducted on large-scale ImageNet to show the effectiveness

**Weaknesses:**

The soundness of this work is quite poor due to the following reasons:

1)	The anti-aliasing and then super-resolution process is conceptually similar to JPEG compression [1], which has been shown to be an unreliable defense method [2]. The improvement of this work is to use the diffusion-based super-resolution method. However, the robustness of diffusion models are also overestimated [3, 4].

2)	No adaptive attacks [2] are evaluated in this work, which also indicates that the results in this paper can be unreliable.

3)	The proposed method is an adversarial detection method. However, no adversarial detection method [5, 6] is compared in this work. The adversarial detection cannot be compared with adversarial defense method directly. The evaluation metric [408-413] can be quite problematic.

Based on these, I think this work should not be published.

[1] Guo C, Rana M, Cisse M, et al. Countering adversarial images using input transformations[J]. arXiv preprint arXiv:1711.00117, 2017.

[2] Athalye A, Carlini N, Wagner D. Obfuscated gradients give a false sense of security: Circumventing defenses to adversarial examples[C] ICML. 2018: 274-283.

[3] Lee M, Kim D. Robust evaluation of diffusion-based adversarial purification[C] ICCV. 2023: 134-144.

[4] Li X, Sun W, Chen H, et al. ADBM: Adversarial diffusion bridge model for reliable adversarial purification[J]. arXiv preprint arXiv:2408.00315, 2024.

[5] Carlini N, Wagner D. Adversarial examples are not easily detected: Bypassing ten detection methods[C]//Proceedings of the 10th ACM workshop on artificial intelligence and security. 2017: 3-14.

[6] Wang Y, Su H, Zhang B, et al. Interpret neural networks by extracting critical subnetworks[J]. IEEE Transactions on Image Processing, 2020, 29: 6707-6720.

**Questions:**

Please see the weaknesses above.

---

### Note · Authors · 2024-11-13

I have read and agree with the venue's withdrawal policy on behalf of myself and my co-authors.